# Investigations into the In Vitro Metabolism of hGH and IGF-I Employing Stable-Isotope-Labelled Drugs and Monitoring Diagnostic Immonium Ions by High-Resolution/High-Accuracy Mass Spectrometry

**DOI:** 10.3390/metabo12020146

**Published:** 2022-02-04

**Authors:** Sophia Krombholz, Andreas Thomas, Mario Thevis

**Affiliations:** 1Institute of Biochemistry/Center for Preventive Doping Research, German Sport University Cologne, Am Sportpark Müngersdorf 6, 50933 Cologne, Germany; s.krombholz@biochem.dshs-koeln.de (S.K.); m.thevis@biochem.dshs-koeln.de (M.T.); 2European Monitoring Center for Emerging Doping Agents (EuMoCEDA), 50933 Cologne, Germany

**Keywords:** peptide metabolism, Insulin-like Growth Factor, growth hormone, high-resolution mass spectrometry, doping

## Abstract

Studying the metabolism of prohibited substances is an essential element in anti-doping research in order to facilitate and improve detectability. Whilst pharmacokinetic studies on healthy volunteers are valuable, they are often difficult, not least due to safety reasons and ethical constraints, especially concerning peptidic substances, which must be administered parenterally. Hence, there is a growing need for suitable in vitro models and sophisticated analytical strategies to investigate the metabolism of protein- and peptide-derived drugs. These include human growth hormone (hGH) and its main mediator insulin-like growth factor-I (IGF-I), both prohibited in professional sports for their anabolic and lipolytic effects, while challenging in their detection, as they occur naturally in the human body.Within this study, the in vitro metabolism of hGH and IGF-I was investigated using a stable-isotope-labelled reporter ion screening strategy (IRIS). A combination of liquid chromatography, high-resolution mass spectrometry, and characteristic immonium ions generated by internal dissociation of the stable-isotope-labelled peptidic metabolites enabled the detection of specific fragments. Several degradation products for hGH and IGF-I were identified within this study. These metabolites, potentially even indicative for subcutaneous administration of the drugs, could serve as promising targets for the detection of hGH and IGF-I misuse in future anti-doping applications.

## 1. Introduction

The continuing progress of modern anti-doping analytics has allowed for great achievements in the last few years, enabling the detection of infinitesimal amounts of prohibited substances and even the establishment of strategies to uncover methods such as gene doping or autologous blood transfusion [1]. These constant enhancements are pivotal to anti-doping efforts. Still, some issues remain challenging, especially including the misuse of substances naturally produced in the human body [1,2,3]. Hence, various procedures have been applied to face this challenge, such as the implementation of threshold levels or isotope-ratio mass spectrometric approaches [4,5,6]. Another important concept is the search for metabolites potentially characteristic of a specific route of administration to discriminate between the endogenous substance and an illicit application of the drug. Therefore, studying the metabolism and excretion pattern of prohibited substances has been a key element of anti-doping research until today [1]. An important tool in this context is the performance of in vitro metabolism studies using human cells, isolated enzymes, or S9 fraction. However, while the analytics of many small molecules in doping control focus on the detection of diagnostic (long-term) metabolites, current strategies applied for the detection of peptides and protein-derived drugs are mostly based on the analysis of the intact drug [7,8]. Advanced analytical techniques, especially liquid chromatography–high-resolution mass spectrometry (LC–HRMS)-based approaches, have enabled a remarkable progress in the investigation of peptide metabolism based on in vitro experiments [9]. In terms of toxicity or bioavailability, this is essential for drug research and development, but it has also led to an increasing number of studies examining in vitro protein and peptide metabolism in the context of anti-doping research, with a particular interest in developing new methods of detecting drug abuse [10,11,12,13,14,15,16]. Peptides included in these studies were, amongst others, insulin, synacthen, and several growth hormone-releasing peptides with, for example, a representative metabolite of insulin generated in vitro being detected within in vivo samples obtained after subcutaneous administration of the drug [12,13,15]. The formation of specific degradation products of peptide hormones during (subcutaneous) administration, and hence the possible detection of the particular fragments in human serum, is a promising approach to identify the misuse of several peptides currently prohibited for performance enhancement in sports, including peptides with anabolic properties such as human growth hormone (hGH) and its main mediator, insulin-like growth factor-I (IGF-I) [17]. The misuse of hGH has frequently been reported for both professional and recreational sports, presumably due to its anabolic and lipolytic effects, as well as the potentially complicated detection in doping controls, considering its natural occurrence in the human body [18,19,20,21]. hGH, also known as somatropin, is a peptide hormone composed of 191 amino acids, and it mainly acts by stimulating the biosynthesis and secretion of IGF-I (primary structures in Figure 1) [22,23,24]. IGF-I consists of 70 amino acids and acts as an anabolic hormone by promoting skeletal growth and protein synthesis in the muscle, as well as increasing glucose and amino acid uptake [25]. Currently, the detection of growth hormone doping in sports is based on two methods: the isoforms differential immunoassay, based on the ratio of recombinant hGH to endogenous hGH, and the hGH biomarkers test, based on the measurement of IGF-I and the N-terminal pro-peptide of collagen type III (P-III-NP) [26,27,28]. Recently, the approach of longitudinal monitoring of hGH biomarkers through the application of the athlete’s biological passport has been proven as a sensitive testing method [29]. However, probing for the abuse of hGH or IGF-I as doping agents remains complex, emphasising the importance of metabolism studies of both peptides, with the potential of complementing and improving existing strategies of growth hormone detection in doping controls.

The identification of metabolites from complex biological matrices, as obtained from in vitro experiments, can be challenging, further complicated by the enormous variety of peptide masses resulting from the degradation of the peptide of interest. A promising approach referred to as isotope-labelled reporter ion screening (IRIS) proved efficient in previous studies to detect metabolites from peptide-derived drugs [15]. In this method, the generation of specific immonium ions from peptides containing stable-isotope-labelled amino acids was used to identify fragments unequivocally resulting from the intact drug. Within this study, the in vitro metabolism of hGH and IGF-I in different biological fluids was investigated by a combination of liquid chromatography–high-resolution mass spectrometry (LC–HRMS) and IRIS, using uniformly stable-isotope-labelled peptides to detect specific metabolites and subsequently resolve their structure via MS/MS-experiments. As there are various proteases involved in the metabolism and degradation of peptides in different tissues, e.g., tryptase and cathepsin G in the skin, S9 fractions of human skin and liver were chosen as incubation media, as well as serum and urine to account for the cell- and tissue-specific distribution of proteolytic enzymes [30,31]. Additionally, the stability of several fragments in human serum was assessed as a first approach to examine their applicability as target analytes in sports drug testing.

**Figure 1 metabolites-12-00146-f001:**
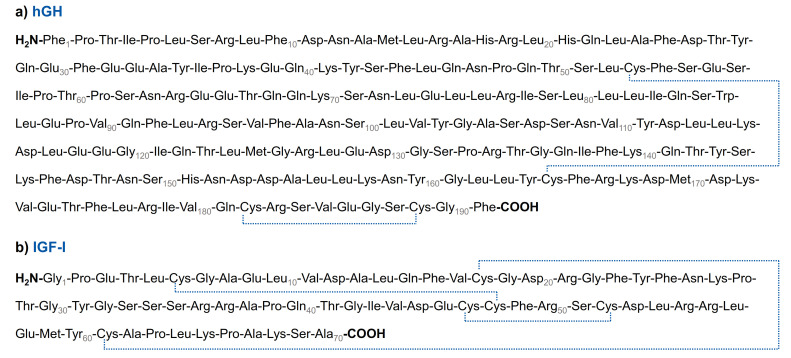
Primary structures of the two peptides examined within this study: (**a**) human growth hormone and (**b**) insulin-like growth factor-I.

## 2. Results and Discussion

The in vitro metabolism of the growth-promoting peptide hormones hGH and IGF-I in different biological fluids was investigated using the presented IRIS approach. Following a simple sample preparation, the survey screen for degradation products of the respective peptide hormone was accomplished by utilising specific immonium ions resulting from the internal dissociation of single ^15^N-labelled amino acids from the peptide backbone as tracer ions. Therefore, mass spectral data were acquired in an all-ion fragmentation mode (AIF), in which all precursor ions in a selected mass range are fragmented by higher energy, collision-induced dissociation before being detected. The collision-induced formation of internal peptide fragments by a combination of *a*-type and *y*-type cleavage produces characteristic signals in the MS/MS spectrum [32]. In Figure 2, the AIF mass spectrum of hGH compared to U-^15^N-labelled hGH in human serum is shown. Several signals matching the immonium ions of “native” amino acids such as Phe (*m*/*z* 120.081) or Tyr (*m*/*z* 136.076) are exhibited in both spectra, whereas in the lower spectrum, the corresponding ^15^N-labelled immonium (“reporter”) ions are additionally visible (^15^N-Phe at *m*/*z* 121.078, ^15^N-Tyr at *m*/*z* 137.073). The natural occurrence of these ions is minimal; therefore, extraction of the particular reporter ion traces leads to diagnostic chromatograms, which enable the identification and, afterwards, the sequencing of the corresponding peptide metabolites.

In a previous study by Thomas et al., this strategy was employed to identify the metabolic products of three model peptides: insulin, synacthen, and corticotropin [15]. These findings demonstrated that a combination of in vitro metabolism studies employing skin S9 microsomes and evaluation of the MS/MS data for stable-isotope-labelled reporter ions represents an effective tool to discover metabolites of peptide-derived drugs. By applying this strategy to hGH and IGF-I using various biological fluids as incubation media, it could be demonstrated that this method is likewise suitable to identify metabolites of larger and more complex structures, and to enable a first comparison between the proteolytic degradation of the peptides in different body tissues. Notably, the uniform labelling of the two peptides presented here enabled a maximised detection efficacy of the tracer ions in comparison to the previous application of IRIS, in which the model compounds only contained a limited number of stable-isotope-labelled amino acids.

**Figure 2 metabolites-12-00146-f002:**
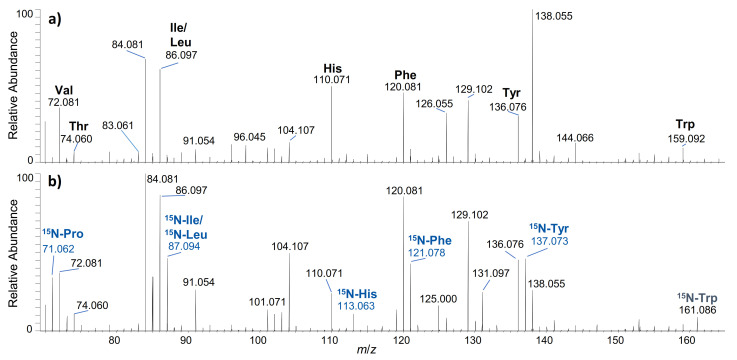
All-ion fragmentation mass spectra depicting the (**a**) non-labelled (“native”) and (**b**) ^15^N-labelled immonium ions generated by collision-induced dissociation of hGH (**a**)) and ^15^N-hGH (**b**)) at the respective retention time.

### 2.1. hGH

For the 22 kDa peptide human growth hormones, extensive metabolism was observed in the skin S9 mix and urine, while fewer fragments were discovered in liver S9 mix and serum. Figure 3 demonstrates the extracted ion chromatograms (EIC) of the most abundant tracer ions of U-^15^N-hGH in skin S9 mix and the associated substrate blank; reporter ions: ^15^N-His (*m*/*z* 113.063), ^15^N-Tyr (*m*/*z* 137.073), ^15^N-Leu/Ile (*m*/*z* 87.094) and ^15^N-Phe (*m*/*z* 121.078). Several additional peaks show the presence of ^15^N-labelled peptide metabolites with relatively high abundances compared to the intact molecule represented by the peak at 9.33 min. Examination of the enzyme blank verified that the degradation occurred due to the presence of proteases in the skin S9 mix and was not solely caused by exposure to temperature and time. Consequently, the full-MS data of all samples were screened for the intact masses and structure determination accomplished by additional MS/MS experiments. In vitro formation of the identified metabolites was then confirmed by the repetition of the experiments with non-labelled native hGH, and the intact hormone could not be detected in any of the blank serum samples. The most abundant metabolites of hGH identified within this study are summarised in Table 1.

In terms of time, most metabolites were detectable after 2 h of incubation, and intensity further increased over time. Compared to the intact peptide, relatively small and predominantly singly or doubly charged fragments were observed, comprising between 6 and 16 amino acids. For two fragments, hGH 29-41 and hGH 29-38, the determined accurate mass suggested an N-terminal cyclisation of glutamine to pyroglutamate (pE). This kind of modification can result both from an enzymatic reaction or from a spontaneous loss of ammonia, depending on the pH and ionic conditions of the sample **[33,34]**. The corresponding product ion mass spectrum of hGH 29-41 is exemplarily depicted in Figure 4, originating from its two-fold charged precursor ion at *m*/*z* 811.391. Accurate masses of the characteristic backbone ions corroborate the sequence and the structure of the N-terminus. Further mass spectra of all presented metabolites are provided in the supplemental information (Appendix A).

As an additional step, the stability of the fragments generated in **the** skin S9 mix was assessed by further incubating the metabolites, generated after 2 h of incubation, in human serum over a period of 8 h. As expected, the metabolic products were subject to further degradation by serum proteases. Only hGH 1-13 and 1-16, representing the two metabolites with an intact N-terminus, could still be detected after 8 h of incubation. These findings might impede the implementation of the metabolites as target analytes to detect GH misuse; however, it is notable that a non-specific sample clean-up was chosen in order to consider all metabolites. An optimised extraction method might enhance the detectability of the peptide fragments and warrants further investigations. The most promising target fragments for doping control analysis are highlighted in Table 1. The bioavailability of subcutaneously administered recombinant hGH is specified between 70 and 90%, indicating that the peptide hormone is partially subject to enzymatic degradation before entering systemic circulation **[35,36,37]**. However, this is affected by the pharmaceutical preparation, dosage, and enzymatic activity, which is why the results of this study do not allow statements concerning the amount of metabolism and the resulting concentration of the peptide fragments and, hence, their detectability in serum. Nevertheless, to the best of our knowledge, this is the first general and systematic study to identify metabolites of hGH in the human body and presents a first step towards implementing potential new target analytes in doping control in order to uncover GH misuse in the future.

### 2.2. IGF-I

The main results on the in vitro metabolism of IGF-I are summarised in Table 2. Altogether, a total number of 9 peptide fragments was successfully identified, with sequences ranging between 8 and 66 amino acids. Figure 5 illustrates the reporter ion trace at *m*/*z* 137.073 (corresponding to the immonium ion of ^15^N-Tyr), extracted from AIF chromatograms after incubation of ^15^N-IGF-I in all biological fluids examined, as well as their respective substrate blanks. This outlines the pronounced formation of metabolites in skin and liver S9 mix, compared to the much lesser extent in serum and urine.

Interestingly and in contrast to hGH, IGF-I was metabolised similarly in skin and liver S9 mix, thus suggesting a similar composition of proteases. Generally, our findings demonstrate a relatively high stability of IGF-I in serum, presumably due to the considerable amount to which the peptide hormone is associated with its IGF-I-binding proteins [38]. As ternary complexes with IGF-I, they play an important role in regulating IGF signalling and increase the half-life of IGF-I. Due to the strong binding, they are also a major issue for the quantification of IGF-I in serum [38,39,40]. Nevertheless, particularly with increasing incubation time, the formation of fragments with a loss of the first two amino acids (IGF-I 3–35 and IGF-I 3–31) could be discovered, as well as the degradation to des(1-3)-IGF-I (IGF-I 4–70), a metabolite already described in previous studies [41,42]. Overall, a predetermined cleavage site appears to be between Pro_2_ and Glu_3_; those metabolites were particularly predominant in the samples incubated over 4–24 h. In Figure 6, the product of the ion mass spectrum of one of the main metabolites formed in skin S9 mix (IGF-I 3–24) is shown, obtained from the two-fold protonated precursor ion at *m*/*z* 1202.551, after reduction with Tris-(2-carboxyethyl)-phosphine hydrochloride (TCEP). All product ion mass spectra of the IGF-I metabolites are presented in the Supplementary Materials (Appendix A)).

In order to consider one or more metabolites of IGF-I as target analytes in doping control, stability is an important aspect; therefore, additional in vitro experiments as described in Section 3.3 were performed with the fragments generated in skin S9 mix. It could be shown that particularly the larger peptide fragments of IGF-I (comprising over 20 amino acids) proved stable after incubation in serum over 8 h. Figure 7 exhibits the EIC of the metabolic products generated in the skin S9 mix spiked in human serum before and after another incubation over a time period of 8 h. The metabolite IGF-I 3–31 could not be detected in serum and is therefore not shown. The smaller peptide fragments of IGF-I generated in the skin S9 mix appear unstable against further enzymatic degradation in serum; however, as mentioned above, detection could be enhanced by an improved clean-up and enrichment strategy in the future. IGF-I concentrations in serum can vary with age, time of the sample draw, and type of assay used. Nevertheless, intact endogenous IGF-I could be detected in all blank serum samples examined within this study. The degradation products identified in vitro that could potentially be formed from endogenous IGF-I in vivo could not be found in any of the blank samples. Based on the findings of these in vitro studies, fragments presenting promising target analytes for doping control analysis are highlighted in Table 2. In the absence of authentic post-administration samples, an assessment as to whether one or more metabolites are detectable in human serum samples after application of recombinant IGF-I (Mecasermin), therapeutically administered in doses up to 0.12 mg/kg of bodyweight twice a day [43], is yet to be completed.

## 3. Materials and Methods

### 3.1. Chemicals and Reagents

For all aqueous buffers and dilution steps, ultrapure water obtained from a Barnstead™ GenPure™ device from Thermo Fisher Scientific (Waltham, MA, USA) was used. Acetic acid, acetonitrile (ACN), formic acid, and phosphate-buffered saline tablets (PBS, pH 7.4) were obtained from Sigma-Aldrich (Schnelldorf, Germany). TCEP was purchased from Carl Roth GmbH (Karlsruhe, Germany). Dimethyl sulfoxide (DMSO) and human liver S9 fraction (total protein concentration of 20 mg/mL) were supplied by Thermo Fisher Scientific (Waltham, MA, USA). Human skin S9 fraction (total protein concentration of 5 mg/mL) was purchased from BioreclamationIVT (Westbury, NY, USA). The reference standards for hGH and U-^15^N-labelled IGF-I were obtained from Prospec (Rehovot, Israel), the respective standard for non-labelled IGF-I from Novozymes (Adelaide, Australia). U-^15^N-labelled 22 kDa hGH was provided by NovoNordisk (thanks to Christian Arsene, Braunschweig, Germany). Serum and urine samples were obtained from healthy human volunteers and stored frozen at −80 °C until usage. The sample collection was approved by the local ethics committee (#139/2021) and all participants provided written consent.

### 3.2. In Vitro Experiments

Different in vitro experiments were performed in order to study the metabolism of hGH and IGF-I. Therefore, 1 µg of the U-^15^N-labelled peptide was incubated in 100 µL of various biological fluids (human skin S9 mix, human liver S9 mix, serum, and urine) under gentle shaking (400 rpm) for 2–24 h at 37 °C. For the S9 mix, 10 µL of the respective S9 fraction was added to 90 µL of PBS. Control experiments without substrates (substrate blank) and without enzymes (enzyme blank) were performed by adding an equal volume of PBS accordingly, in order to differentiate the generated metabolites from matrix-associated artefacts. Afterwards, 50 µL of CH_3_COOH 2% and 400 µL of ACN were added to the samples and the precipitated proteins separated by centrifugation at 17,000× *g* for 10 min. The supernatant was transferred into clean Eppendorf Lo-Bind tubes prior to being concentrated by vacuum-centrifugation. The dry residue was reconstituted in 100 µL of aqueous CH_3_COOH 1% and, after another centrifugation, ready for injection. In addition, all experiments were performed identically with the non-labelled peptides to ensure the transferability of the results to authentic samples. To facilitate structure determination, the samples were further incubated with TCEP solution (100 mM in CH_3_COOH 1%) at 37 °C for 1 h to reduce intramolecular cysteine–cysteine bonds.

### 3.3. Fragment Stability in Human Serum

As stability of the peptide metabolites in serum is crucial in order to consider them as potential target analytes, further degradation of the metabolites generated by incubation in skin S9 mix, mimicking subcutaneous administration, was assessed by additional in vitro experiments. Therefore, 20 µL of the skin S9 in-vitro incubation solution from above (obtained after 2 h, reconstituted in H_2_O) was spiked into 100 µL of human serum. This fortified specimen was measured both before and after 8 h of incubation at 37 °C under gentle shaking, following sample preparation as described above. The same experiments were performed with the respective substrate blanks.

### 3.4. Liquid Chromatography–High-Resolution Mass Spectrometry

Separation and detection of the peptide fragments were accomplished using LC-HRMS, consisting of a Thermo Fisher Scientific Vanquish™ Duo UHPLC system coupled to an Orbitrap Exploris™ 480 mass spectrometer by Thermo Fisher Scientific (Bremen, Germany). Eluent A was composed of formic acid 0.1%/DMSO 1% in water and eluent B of formic acid 0.1%/DMSO 1% in ACN. Prior to the analytical column, the samples were first trapped for 2 min. on an Accucore Phenyl-Hexyl column (3 × 10 mm, 2.6 µm, Thermo Fisher Scientific, Bremen, Germany) with a flow rate of 400 µL/min. and 99% eluent A. Afterwards, the flow was directed to a Poroshell EC-C18 column (3 × 50 mm, 2.7 µm, Agilent Technologies, Santa Clara, CA, USA). With a total run time of 15 min., the gradient started with 1% B for 2 min. (trapping), followed by an increase to 40% B within 6 min., then increased to 80% B in 2 min. before re-equilibrating at the starting conditions of 1% B. The injection volume was 10 µL, and the flow rate was set at 400 µL/min. The MS was equipped with a heated electrospray ion source operating in positive mode with an ionisation voltage of 3000 V. The temperature of the ion transfer capillary was set at 320 °C. For monitoring of the stable-isotope-labelled reporter ions and subsequent detection of the peptide fragments, all-ion fragmentation experiments (AIF) were performed, with an isolation range of *m*/*z* 800–2000 and a scan range of *m*/*z* 70–700 (resolution 60,000 FWHM). The normalised collision energy (NCE) was set at 30%. For hGH, the isolation window was adapted to *m*/*z* 800–2500. Further, full-scan data were acquired in a scan range of *m*/*z* 400–1700 for IGF-I and *m*/*z* 550–2100 for hGH, accordingly (resolution 60,000 FWHM), in order to screen for the intact masses of the detected metabolites. For structure elucidation and amino acid sequencing, MS/MS experiments with an isolation window of *m*/*z* 2 and individually optimised collision energies were performed (resolution 30,000 FWHM). Main LC-MS characteristics of the target analytes are summarised in Table 1 and Table 2.

## 4. Conclusions

Investigating the metabolism of peptide- and protein-derived drugs presents a challenge in anti-doping research and analytical chemistry. Within this study, it could be demonstrated that the combination of in vitro metabolism and the application of mass spectrometric analysis based on stable-isotope-labelled tracer ions enabled the characterisation of metabolites of two important peptide hormones: human growth hormone and insulin-like growth factor-I. For both peptides, the formation of a variety of different metabolites in various biological fluids could be shown and the corresponding structures determined. The extensive degradation of the peptides in the skin S9 mix provides a promising basis to potentially use these fragments to uncover the subcutaneous administration of the recombinant peptide hormones, albeit most of the smaller fragments seem to be subject to further degradation in human serum. Future work should be devoted to optimising the detection of the metabolites in serum, and subsequently examine authentic post-administration samples to contribute to improving the analytics of hGH and IGF-I, not only in doping controls. Overall, this study demonstrates the applicability of the presented IRIS approach to important and structurally complex peptides, substantiating how this tool can be used to explore the metabolism of further peptide-based drugs: an essential task for doping analysis and clinical research in general.

## Figures and Tables

**Figure 3 metabolites-12-00146-f003:**
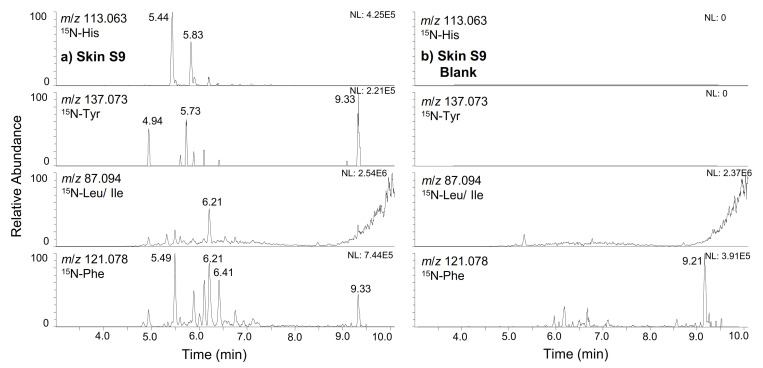
AIF chromatogram of the extracted ion traces corresponding to the most abundant ^15^N-labelled immonium ions, showing the metabolites obtained after incubation of ^15^N-hGH in skin S9 mix (**a**)) in comparison to the respective substrate blank (**b**)).

**Figure 4 metabolites-12-00146-f004:**
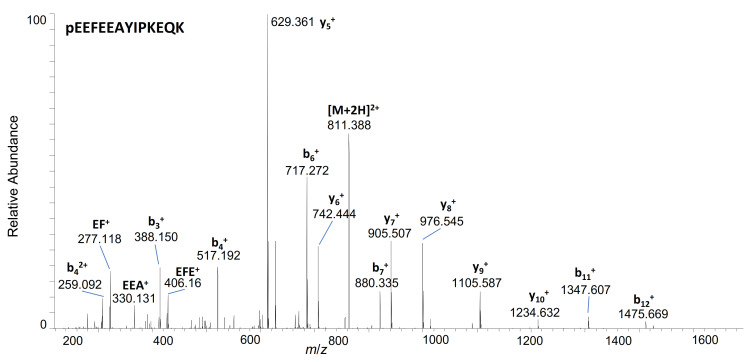
Product ion mass spectrum of unlabelled hGH 29-41 obtained after incubation of hGH in skin S9 mix for 24h; precursor selected at m/z 811.391 (z = 2).

**Figure 5 metabolites-12-00146-f005:**
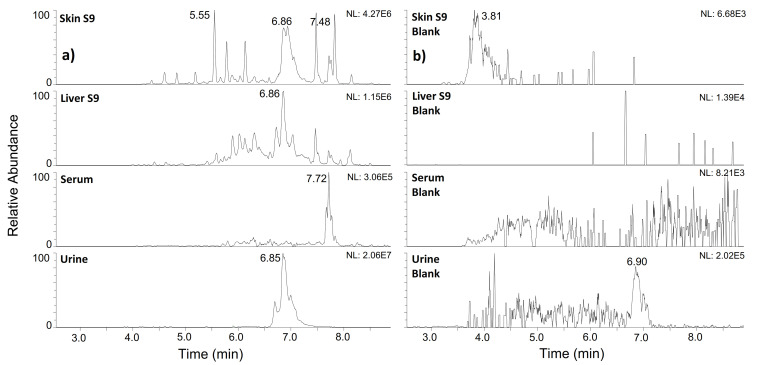
AIF chromatogram extracted at *m*/*z* 137.073 corresponding to the ^15^N-Tyr immonium ion obtained after incubation of ^15^N-IGF-I in different biological fluids (**a**)) compared to the respective substrate blanks (**b**)).

**Figure 6 metabolites-12-00146-f006:**
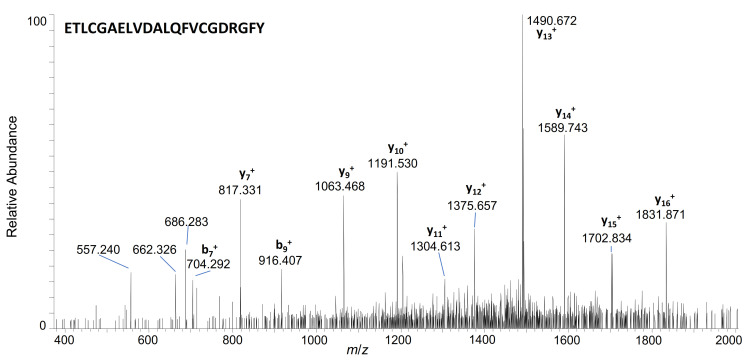
Product ion mass spectrum of unlabelled IGF-I 3–24, generated by incubation of IGF-I in skin S9 mix over 24 h; precursor selected at *m*/*z* 1202.551 (z = 2).

**Figure 7 metabolites-12-00146-f007:**
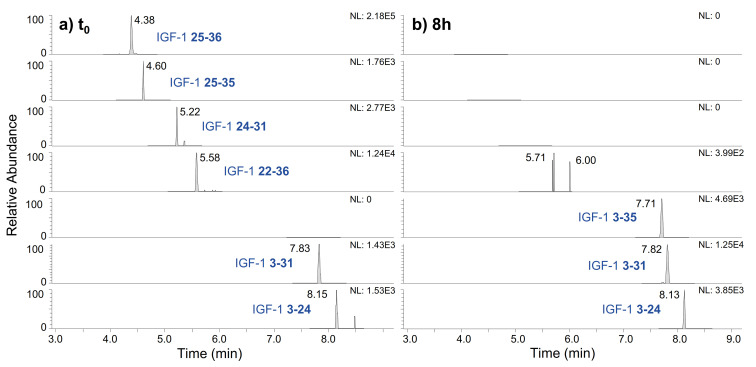
Extracted ion chromatograms of IGF-I metabolites generated by incubation in skin S9 mix over 2 h (**a**) before and (**b**) after another incubation in human serum over 8 h to assess stability.

**Table 1 metabolites-12-00146-t001:** Amino acid sequences and main characteristics of the metabolites derived from hGH using the presented IRIS approach. The most promising fragments for doping control analysis are highlighted in bold. (+ indicates the detection).

Peptide	Amino Acid Sequence	Monoisotopic Mass (Da)	Dominant Charge State	Retention Time (min)	Product of Proteolytic System			
					Skin S9	Liver S9	Serum	Urine
Intact	-	1106.5593	20+	9.21	+	+	+	+
135–142	TGQIFKQT	922.4993	1+	4.96	+			
135–143	TGQIFKQTY	1085.5626	1+	5.48	+	+		
34–44	AYIPKEQKYSF	687.3586	2+	5.52	+			+
125–139	MGRLEDGSPRTGQIF	832.4147	2+	5.86				
**29–41**	**pE * EFEEAYIPKEQK**	811.3909	2+	5.89	+			
**9–16**	**LFDNAMLR**	979.5030	1+	6.21	+			
86–91	WLEPVQ	771.4036	1+	6.25	+	+		+
102–113	VYGASDSNVYDL	1302.5848	1+	5.32				+
114–124	LKDLEEGIQTL	629.8481	2+	6.36				+
29–38	pE*EFEEAYIPK	1236.5783	1+	6.41	+		+	
86–92	WLEPVQF	918.4720	1+	7.45	+			+
**1–13**	**FPTIPLSRLFDNA**	745.9037	2+	7.80	+			+
1–16	FPTIPLSRLFDNAMLR	631.0135	3+	8.20	+		+	

* pE: pyroglutamate.

**Table 2 metabolites-12-00146-t002:** Amino acid sequences and main characteristics of the metabolites derived from IGF-I using the presented IRIS approach. The most promising fragments for doping control analysis are highlighted in bold. (+ indicates the detection).

**Peptide**	Amino Acid Sequence	Monoisotopic Mass (Da)	Dominant Charge State	Retention Time (min)	Product of Proteolytic System			
					Skin S9	Liver S9	Serum	Urine
Intact		956.9592	8+	6.89	+	+		+
25–36	FNKPTGYGSSSR	650.8177	2+	4.36	+	+		+
25–35	FNKPTGYGSSS	572.7671	2+	4.60	+	+		+
24–31	YFNKPTGY	989.4727	1+	5.18	+			
**22–36**	**GFYFNKPTGYGSSSR**	834.3943	2+	5.55	+	+		+
4–70	TLCGAELVDALQFVCGDRGFYFNKPTGYGSSSRRAPQTGIVDECCFRSCDLRRLEMYCAPLKPAKSA	921.0660	8+	6.85	+	+	+	+
3–36	ETLCGAELVDALQFVCGDRGFYFNKPTGYGSSSR	922.1817	4+	7.48	+	+		
3–35	ETLCGAELVDALQFVCGDRGFYFNKPTGYGSSS	1177.2061	3+	7.72	+	+	+	
3–31	ETLCGAELVDALQFVCGDRGFYFNKPTGY	1071.167	3+	7.83	+	+	+	
**3–24**	**ETLCGAELVDALQFVCGDRGFY**	1202.5510	2+	8.15	+	+		

## Data Availability

Data is contained within the article or supplementary material..

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
