# Peer review of "Investigations into the In Vitro Metabolism of hGH and IGF-I Employing Stable-Isotope-Labelled Drugs and Monitoring Diagnostic Immonium Ions by High-Resolution/High-Accuracy Mass Spectrometry"

_metabolites, 2022, doi:10.3390/metabo12020146_

Round 1

Reviewer 1 Report

Sophia Krobholz et al. present a really well-written manuscript regarding the metabolism of human hGH and IGF-I serving for the future development of doping control assays. The research question is outlined in a clear fashion, all methods are very well described and the results presented in a very comprehensible fashion, I really appreciate the professional style of the manuscript. The following aspects may help to further improve the manuscript, which otherwise fulfills all requirements for publication.

  • the use of S9 microsome fractions is common for toxicologists to investigate catabolism. However, this applies to small molecules and toxins rather than peptides and proteins. Here, lysosomal proteases might be even more important. I understand this will be contained in the s9 fraction as well, but it might help to introduce these implications with some more details such as listing candidate proteases
  • the properties and implications of AIF might not be obvious for all reades who are not that familiar with MS and should also be introduced with a few words.

Author Response

Please find below the original comments of reviewer 1 and our actions marked with an asterisk:

Reviewer 1:

Sophia Krombholz et al. present a really well-written manuscript regarding the metabolism of human hGH and IGF-I serving for the future development of doping control assays. The research question is outlined in a clear fashion, all methods are very well described and the results presented in a very comprehensible fashion, I really appreciate the professional style of the manuscript. The following aspects may help to further improve the manuscript, which otherwise fulfills all requirements for publication.

the use of S9 microsome fractions is common for toxicologists to investigate catabolism. However, this applies to small molecules and toxins rather than peptides and proteins. Here, lysosomal proteases might be even more important. I understand this will be contained in the s9 fraction as well, but it might help to introduce these implications with some more details such as listing candidate proteases

*To explain the application of S9 fraction, the following sentence was added in the introduction: “As there are various proteases involved in the metabolism/ degradation of peptides in different tissues, e.g. tryptase and cathepsin G in the skin, S9 fractions of human skin and liver were chosen as incubation media, as well as serum and urine to account for the cell- and tissue-specific distribution of proteolytic enzymes.”

the properties and implications of AIF might not be obvious for all reades who are not that familiar with MS and should also be introduced with a few words.

*The following sentence was added in the results section: “Therefore, mass spectral data was acquired in “all ion fragmentation” mode (AIF), where all precursor ions in a selected mass range are fragmented by higher energy collision-induced dissociation before being detected. “

Reviewer 2 Report

The paper is of interest and it is well presented and structured.

The paper expands previous experience applied to insulin and Synacthen  (see reference 15 of the manuscript) to GH and IGF-1. The introduction is academically correct but probably too long before reaching the real aims of the work. A reduction of the introduction focussing on the goals of the work is strongly suggested.

In the experimental part, the incubations with serum are presented. Data on products of proteolytic conditions are presented in tables 1 and 2, but urine is not mentioned in materials and methods.  The scope of the stability of the S9 incubations in serum is clearly presented there but not during the discussion of the results. This should be reviewed for clarification.

In the supplementary material for hGH fragment 135-142  [M+2H]2+ is shown while in table 1 one charge is indicated. Also in the supplementary material, some decimal points of the masses are indicated with a hyphen (-) and not a dot (.), please check.

Author Response

Please find attached the original comment of reviewer 2 and our actions answer markes with an asterisk:

Reviewer 2:

The paper is of interest and it is well presented and structured.

The paper expands previous experience applied to insulin and Synacthen  (see reference 15 of the manuscript) to GH and IGF-1. The introduction is academically correct but probably too long before reaching the real aims of the work. A reduction of the introduction focussing on the goals of the work is strongly suggested.

*Several parts of the introduction concerning the pharmacology of hGH and IGF-I were removed, to set a stronger focus on the goals of this study.

In the experimental part, the incubations with serum are presented. Data on products of proteolytic conditions are presented in tables 1 and 2, but urine is not mentioned in materials and methods.  The scope of the stability of the S9 incubations in serum is clearly presented there but not during the discussion of the results. This should be reviewed for clarification.

*The incubation in human urine is mentioned in the material and methods, page 9, line 270. The results of the stability study were included in the results& discussion section of each peptide, as they were directly related to the results of the in vitro experiments themselves. To highlight the findings concerning the stability, an introducing sentence was added for IGF-I: “In order to consider one or more metabolites of IGF-I as target analytes in doping control, stability is an important aspect, therefore additional in vitro experiments as described in section 4.3 were performed with the fragments generated in skin S9 mix.I could be shown that particularly…”. In the section about hGH, a similar introduction already existed.

In the supplementary material for hGH fragment 135-142  [M+2H]2+ is shown while in table 1 one charge is indicated. Also in the supplementary material, some decimal points of the masses are indicated with a hyphen (-) and not a dot (.), please check.

*The figures and figure captions were changed accordingly.
